# Uptake of Sulfate from Ambient Water by Freshwater Animals

**Michael B. Griffith [1,2,\*]** , **James M. Lazorchak [2,3]** and **Herman Haring [4]**

1    U.S. Environmental Protection Agency, Office of Research and Development, National Center for Environmental Assessment, Cincinnati, OH 45268, USA

2    U.S. Environmental Protection Agency, Office of Research and Development, Center for Environmental Measurement and Modeling, Cincinnati, OH 45268, USA; lazorchak.jim@epa.gov

3    U.S. Environmental Protection Agency, Office of Research and Development, National Exposure Research Laboratory, Cincinnati, OH 45268, USA

4    Pegasus Technical Services, Inc., Cincinnati, OH 45268, USA; herman.haring@hotmail.com

\*    Correspondence: griffith.michael@epa.gov

**Abstract:** To better understand how the sulfate ($SO_4^{2-}$) anion may contribute to the adverse effects associated with elevated ionic strength or salinity in freshwaters, we measured the uptake and efflux of $SO_4^{2-}$ in four freshwater species: the fathead minnow (*Pimephales promelas*, Teleostei: Cyprinidae), paper pondshell (*Utterbackia imbecillis*, Bivalvia: Unionidae), red swamp crayfish (*Procambarus clarkii*, Crustacea: Cambaridae), and two-lined mayfly (*Hexagenia bilineata*, Insecta: Ephemeridae). Using $\delta(^{34}S/^{32}S)$ stable isotope ratios and the concentrations of S and $SO_4^{2-}$, we measured the $SO_4^{2-}$ influx rate ($J_{in}$), net flux ($J_{net}$), and efflux rate ($J_{out}$) during a 24 h exposure period. For all four species, the means of $J_{in}$ for $SO_4^{2-}$ were positive, and $J_{in}$ was significantly greater than 0 at both target $SO_4^{2-}$ concentrations in the fish and mollusk and at the lower $SO_4^{2-}$ concentration in the crayfish. The means of $J_{out}$ and $J_{net}$ were much more variable than those for $J_{in}$, but several species by target $SO_4^{2-}$ concentration combinations for $J_{out}$ and $J_{net}$, were negative, which suggests the net excretion of $SO_4^{2-}$ by the animals. The results of our experiments suggest a greater regulation of $SO_4^{2-}$ in freshwater animals than has been previously reported.

**Keywords:** sulfate; freshwater; uptake; efflux; fish; invertebrates

## 1. Introduction

Water quality benchmarks were recently developed for elevated total ion concentrations in freshwaters using specific conductance as a measurement endpoint to protect aquatic life [1,2]. However, there is still uncertainty about how different ions contribute to the adverse effects associated with elevated total ionic strength or salinity on freshwater biota, and some studies have suggested a more traditional approach where the concentrations of specific anions should be the targets of chemical monitoring and ambient water quality criteria [3–6]. The predominant form of dissolved sulfur (S) in water is the anion, sulfate ($SO_4^{2-}$), which can be especially elevated in waters affected by mining. Mining exposes sulfide-rich minerals, such as pyrite associated with coal mining and other metal sulfides associated with hard rock metal mining [7–10]. Under oxic conditions, these sulfides are rapidly transformed into $SO_4^{2-}$.

Several studies have recently attempted to assess the ecotoxicity of $SO_4^{2-}$ with standard bioassays that have exclusively used $Na_2SO_4$ [11–14]. The objective of at least some of these studies is the compilation of sufficient bioassay data for a species sensitivity distribution to develop an ambient water criterion for $SO_4^{2-}$. These studies attribute many of the observed adverse effects to $SO_4^{2-}$. This

assumes that any effects associated with the concurrently elevated concentrations of $Na^+$ are minor, even though the uptake of $Na^+$ across epithelial membranes is well known [15–17]. Additionally, the uptake of $SO_4^{2-}$ across external epithelial membranes, such as gills, occurs and has an ionoregulatory or osmoregulatory adverse effect on these animals. However, because the transport physiology of $SO_4^{2-}$ in freshwater animals is relatively unstudied, the validity of the second assumption is unknown. Other studies have begun to study the interactions between different anion and cation combinations [18–20].

To support decisions relating to the potential for adverse effects associated with individual major ions, $Na^+$, $K^+$, $Ca^{2+}$, $Mg^{2+}$, $Cl^-$, $SO_4^{2-}$, and $HCO_3^-$, we previously published a literature review on the ion physiology of freshwater species of four animal groups: teleost fish, crustaceans, aquatic insects, and mollusks [21]. While this review found extensive literature on many of these major ions at least for teleost fish, if not always for the freshwater invertebrates, it identified a data gap related to the ionoregulatory physiology of $SO_4^{2-}$. Limited data appear to suggest that $SO_4^{2-}$ is relatively impermeant to the gill membranes of fish [22,23], crustaceans [24,25], or unionid mussels [26]. However, $SO_4^{2-}$-transporters are present in amphibian skin [27], which functions in ion uptake similarly to the gill membranes of fish, crustaceans, and mollusks or to anal papillae, chloride epithelia, or other epithelial surfaces in aquatic insects [21]. However, see the research on $SO_4^{2-}$ uptake from the water that has been recently published on aquatic insects by Scheibener et al. [28] and Buchwalter et al. [29].

Like other major ions, $SO_4^{2-}$ is present in the extracellular fluids (i.e., blood or hemolymph) of freshwater animals. Reported $SO_4^{2-}$ concentrations include 0.76 ± 0.18 mmol/L (n = 6); 2.58 ± 0.06 (n = 30) and 2.60 ± 0.90 mmol/L (n = 7) in two species of *Anodonta* and zebra mussels (*Dreissena polymorpha*), respectively (Mollusca) [30–32]; 0.94 ± 0.26 for the spinycheek crayfish (*Orconectes limosus*) [33]; and 0.89 and 2.14 mmol/L for white sucker (*Catostomus commersoni*) and rainbow trout (*Oncorhynchus mykiss*), respectively [34]. A greater $SO_4^{2-}$ concentration of 18.7 mmol/L was observed in freshwater Japanese eels (*Anguilla japonica*) [35], but the authors suggest this catadromous species balances greater $SO_4^{2-}$ with less $Cl^-$ in its blood.

Moreover, $SO_4^{2-}$ is metabolically active. S is a component of the amino acids methionine and cysteine, which produce S-containing proteins, and a component of other biomolecules, such as sulfated glycosaminoglycans, which are generally ubiquitous in metazoans, including mollusks, crustaceans, insects, and fish [36–39]. In most organisms, the initial step in S assimilation is $SO_4^{2-}$ activation to 3'-phosphoadenosine 5'-phosphosulfate (PAPS) [40–43].

Freshwater animals are generally hyperregulators that maintain greater ion concentrations in their blood or hemolymph than are found in surrounding freshwaters [44]. As the external medium is hypoosmotic or more dilute than body fluids, these species deal with the continuous diffusional loss of salts and the osmosis of water across their permeable membranes. Therefore, water balance is accomplished by the excretion of dilute waste fluids by their renal systems. Salt concentrations are maintained by the function of various ion transporting proteins in epithelial membranes, such as in the gills, gastrointestinal system, or renal system, that allow the active transport of ions against concentration gradients (i.e., the absorption of ions from the surrounding water or food or the reabsorption of ions from waste fluids, such as urine). Furthermore, most freshwater species, unlike saltwater species, limit their drinking of water, thereby limiting the absorption of water through the gastrointestinal system and dilution of the hemolymph. However, increased water concentrations of some ions, such as $SO_4^{2-}$, change concentration gradients across the epithelial membranes involved in ionoregulation, such as the gills. This may change $SO_4^{2-}$ influx across these membranes, increase blood or hemolymph $SO_4^{2-}$ concentrations, and have osmoregulatory or other adverse effects. However, it is also possible that these epithelial membranes could be impermeant to $SO_4^{2-}$, and none of the above effects may occur. To test whether $SO_4^{2-}$ can move across epithelial membranes, we conducted laboratory experiments with a fish, a crustacean, a mollusk, and a mayfly. These experiments used protocols similar to traditional toxicity tests in that the organisms were exposed to reconstituted water with elevated $SO_4^{2-}$ concentrations. However, the concentrations were not expected to have overtly adverse effects, such as mortality. Moreover, the reconstituted water was made, in part, with enriched

Na$_2$[$^{34}$S]O$_4$ to elevate its $\delta(^{34}S/^{32}S)$. After exposure, the whole body stable isotope ratios of S were measured to assess whether the $^{34}S$ associated with animal tissues increased.

This study was designed to fill a data gap by conducting laboratory experiments with species from four freshwater animal groups (crustaceans, fish, unionid mussels, and aquatic insects) to measure SO$_4{}^{2-}$ uptake from ambient waters and the effect of ambient SO$_4{}^{2-}$ concentrations on SO$_4{}^{2-}$ uptake. Significant SO$_4{}^{2-}$ uptake from the water into an animal would show that the epithelial membranes are permeant to SO$_4{}^{2-}$, likely via an ion-specific transporter. This SO$_4{}^{2-}$ uptake would affect internal SO$_4{}^{2-}$ concentrations, would play a direct role in ionoregulation by the animal, and could have adverse effects if elevated in freshwaters.

## 2. Materials and Methods

To test whether SO$_4{}^{2-}$ uptake is similar or differs among different major taxa groups of freshwater animals, we conducted parallel experiments using a representative species in each of four taxa groups. The ion transport physiology of these species is unlikely to change with the aquatic developmental stage [21]. Therefore, juvenile fish and crayfish, adult unionid mussels, and later instar mayfly nymphs were used in the experiments. The exposures for each of the four species was conducted during a single 24 h period, but the experiments were conducted at different times between September 2017 and April 2019.

### 2.1. Test Animals

Although we considered, originally, using model species (e.g., Cladocera and Chironomidae), often used by toxicity studies, our methods required larger individuals to provide sufficient biomass for stable isotope analysis. We selected the fathead minnow (*Pimephales promelas* Rafinesque, 1820) which is commonly used as a model for a teleost fish. Less conventional test species included a mollusk, the paper pondshell (*Utterbackia imbecillis* (Say, 1829)); a crustacean, the red swamp crayfish (*Procambarus clarkii* (Girard, 1852)); and an aquatic insect, the two-lined mayfly (*Hexagenia bilineata* (Say, 1824)).

We used fathead minnows from laboratory colonies usually cultured and maintained for toxicity testing according to [45,46]. We fed them live *Artemia* nauplii twice a day at a rate of 1 mL per 20 L tank per day-of-age per feeding with a nauplii suspension of 15 mL of brine shrimp cysts (Brine Shrimp Direct, Ogden, UT, USA) incubated for 24 h at 28 °C in aerated Labline water with 25 mL NaCl added.

We purchased red swamp crayfish from Carolina Biological Supply Co., Burlington, NC, USA. The crayfish were placed in a tank with about 175 L of water aerated with air stones. Lengths of polyvinyl chloride pipe cut in half lengthwise were placed in the aquaria to provide cover for the crayfish and reduce aggression. The crayfish were fed thawed adult *Artemia* ad libitum daily.

We obtained the paper pondshells from the Kentucky Department of Fish and Wildlife Resources' Center for Mollusk Conservation (Frankfort, KY, USA). The mussels were held in aquaria, with 28 individuals per aquarium, where the water was aerated by an air stone. The mollusks were fed 20 mL of FFAY, an internally made mixture of fish flake food (Tetramin®, Tetra, Blacksburg, VA, USA), alfalfa (from capsule; Nature's Way, Fargo, ND, USA), and yeast (Flieschmann's, Oakbrook Terrace, IL, USA); and 15 mL of an algal culture of flagellated algae and diatoms (Shellfish Diet 1800®, Reed Mariculture, Campbell, CA, USA) and 15 mL of alfalfa per tank daily.

We purchased the two-lined mayfly from The Reel Thing Live Bait, Green Bay, WI, USA. The mayfly nymphs were held in aquaria, where the water was aerated by an air stone, and were fed ground cereal grain flake food (Cerophyll®, Ward's Natural Science Establishment, Inc., Rochester, NY, USA).

Before their use in experiments, we held the fish and invertebrates in dechlorinated and hardness-adjusted municipal tap water (mean ionic composition in mmol L$^{-1}$: 1.04 [Na$^+$], 0.05 [K$^+$], ~0.25–0.55 [Ca$^{2+}$], 0.43 [Mg$^{2+}$], $\leq$ 0.01 [Cl$^-$], and 0.93 [SO$_4{}^{2-}$]; pH: 7.2–8.1; total organic carbon (TOC): ~0.08 mg C L$^{-1}$; and hardness: ~150–180 mg L$^{-1}$ as CaCO$_3$). All animals, except the two-lined

mayflies, were maintained at a constant temperature of 25 ± 1 °C with a 16:8 h light to dark photoperiod. The two-lined mayflies were maintained at a constant temperature of 15 ± 1 °C.

For at least 24 h before the individual experiments, the animals were acclimated to a modified moderately hard reconstituted water (MMHRW) similar to the R-MHRW of Smith et al. [47] generated by the addition of reagent-grade salts to deionized water produced by a Millipore Super-Q Plus water purification system and bubbled with $CO_2$ to dissolve the $CaCO_3$, particularly using $Na_2SO_4$ (Table 1). During the 24 h acclimation and test phases, the animals were not fed, while the holding water temperatures were maintained.

**Table 1.** Concentrations of salts in the modified moderately hard reconstituted water (MMHRW) used in the experiment along with the resulting target mmol $L^{-1}$ of the major ions, along with hardness and the molar $[Na^+]/[K^+]$ ratios. Only the concentrations of $Na^+$ and $SO_4^{2-}$ were increased in the MMHRW.

| Water | Salt | mg $L^{-1}$ | Ions | mmol $L^{-1}$ |
|---|---|---|---|---|
| Acclimation | $CaCO_3$ | 71.0 | $Na^+$ | 0.54 |
| | $NaHCO_3$ | 4.0 | $K^+$ | 0.05 |
| | $MgCl_2·6H_2O$ | 59.4 | $Ca^{2+}$ | 0.71 |
| | KCl | 3.5 | $Mg^{2+}$ | 0.29 |
| | $Na_2SO_4$ | 35.0 | $HCO_3^-$ | 1.47 |
| | | | $Cl^-$ | 0.34 |
| | | | $SO_4^{2-}$ | 0.25 |
| | | | Hardness (mg/L, $CaCO_4$) | 100.2 |
| | | | Molar $[Na^+]/[K^+]$ | 11.5 |
| 0.49 mmol $L^{-1}$ $SO_4^{2-}$ | $Na_2SO_4$ | 70.0 | $Na^+$ | 1.03 |
| | | | $SO_4^{2-}$ | 0.49 |
| | | | Hardness (mg/L, $CaCO_4$) | 100.2 |
| | | | Molar $[Na^+]/[K^+]$ | 22.0 |
| 1.23 mmol $L^{-1}$ $SO_4^{2-}$ | $Na_2SO_4$ | 175.0 | $Na^+$ | 2.51 |
| | | | $SO_4^{2-}$ | 1.23 |
| | | | Hardness (mg $L^{-1}$, $CaCO_4$) | 100.2 |
| | | | $[Na^+]/[K^+]$ | 53.5 |

MMHRW = modified moderately hard reconstituted water.

## 2.2. Test Water

Upon the initiation of the experiment, the animals were exposed to two concentrations of $SO_4^{2-}$, with the MMHRW with $Na_2SO_4$ added to increase by 2× (i.e., 0.49 mmol $L^{-1}$) or 5× (1.23 mmol $L^{-1}$) the $SO_4^{2-}$ concentrations to test whether differing water $SO_4^{2-}$ concentrations influenced uptake (Table 1). This addition also increased $Na^+$ concentrations to 1.03 mmol $L^{-1}$ and 2.51 mmol $L^{-1}$, respectively. None of these concentrations are near the concentrations that have been observed to have acute adverse effects on freshwater animals [6,11,12,48]. Moreover, the $Na^+/K^+$ molar ratios were 22.0 and 53.5, respectively, which are within the range where alterations of this ratio have not been observed to have osmoregulatory effects [49,50].

To observe influx rates, test water made with unenriched reagent grade $Na_2SO_4$ was mixed with the same concentration test water made with 90% atom enriched $Na_2[^{34}S]O_4$ (Sigma-Aldrich, Miamisburg, OH, USA) at a ratio of 939 parts of reagent grade test water to 61 parts of 90% atom enriched test water. A test water made with only 90% atom enriched $Na_2[^{34}S]O_4$ could not be used without dilution to reduce the enrichment of the $^{34}S$. Too strong of a signal from $^{34}S$ when the samples are analyzed in the mass spectrometer can overload the detectors and cause falsely elevated concentration readings in subsequent samples (R. Venkatapathy, personal communication).

### 2.3. Sampling Methods

#### 2.3.1. Measurements of Animals and Water

Measurements were made on organisms placed in containers with aeration, containing 200 mL, 12 L, 7 L, and 150 mL of the waters for the minnow, unionid mollusk, crayfish, and mayfly, respectively. The volumes were chosen based on the mass loading limits for the animals in static tests [45]. The two-lined mayflies were supplied with short lengths (~38 mm) of 9.5 mm inner-diameter vinyl tubing as artificial burrows [51]. In all four species, the mass of the animal (i.e., ≥10 mg dry mass) was sufficient for chemical analysis, permitting us to measure each individual.

Subsamples of the test waters were taken to measure the $SO_4^{2-}$ concentrations and $\delta(^{34}S/^{32}S)$ for each exposure at 0 h, and a water sample was taken from each exposure container to measure the $SO_4^{2-}$ concentrations after the 24 h exposures. At least 10 extra individuals of each species were sacrificed at 0 h to measure initial whole-body S concentrations and $\delta(^{34}S/^{32}S)$. Measurements from 25 separate replicates of each $SO_4^{2-}$ concentration were collected for each test species. From every exposure, replicate whole-body samples of each test species were taken at the end of the exposure to the enriched stable isotope water and used to measure whole-body S concentrations and $\delta(^{34}S/^{32}S)$. However, because there was some mortality in the two-lined mayfly experiment, the numbers of replicates were 17 and 22 for the 0.49 and 1.25 mmol L$^{-1}$ $SO_4^{2-}$ exposures, respectively. Moreover, an error in the initial water $SO_4^{2-}$ analysis in the mayfly experiment reduced the number of valid measurements of the initial 0.49 mmol L$^{-1}$ water $SO_4^{2-}$ concentration to 1, precluding valid means of $J_{out}$ and $J_{net}$ for the 0.49 mmol L$^{-1}$ $SO_4^{2-}$ exposure.

At the end of each exposure, the fish, crayfish, or mayfly nymph individuals were removed from the containers, rinsed in deionized water to remove any stable isotope label from the surface, and blotted dry on filter paper. For the unionid mussel, the soft tissue was cut from the shell. The fish, mussels (soft tissue only), and mayfly nymphs were dried overnight at 90 °C, whereas the crayfish were freeze-dried for 14 days to weaken their exoskeletons. The animals were then weighed, homogenized, and powdered to 100–200 μm with a mortar and pestle, stored in vials, and analyzed for total S and $\delta(^{34}S/^{32}S)$ [52].

#### 2.3.2. Chemical Analysis of Test Water and Animals

The total concentrations of $SO_4^{2-}$ in the test waters were measured with ion chromatography (EPA Method 300.0) [53]. For the stable isotope analysis of $SO_4^{2-}$ in the initial test waters, we precipitated the dissolved $SO_4^{2-}$ in subsamples of the water placed in the test chambers as barium sulfate (BaSO$_4$) following Révész et al. [54].

The total S concentration of the dried animal tissues was measured using an elemental analyzer. The molar ratios of $^{34}S/^{32}S$, reported as the deviation of this ratio from the international standard, Canon Diablo Troilite, or $\delta(^{34}S/^{32}S)$ [55], were measured in both the BaSO$_4$ precipitates and dried animal tissues with an elemental analyzer connected to a continuous-flow 20-20 gas source stable isotope ratio mass spectrometer [52,54]. For $^{34}S$ analyses, the sample was placed in a tin capsule with vanadium dioxide for combustion, the combustion reactor was held at 1080 °C, and the reaction tube contained tungsten oxide on alumina as an oxidative catalyst and copper metal to remove excess O$_2$ subsequent to combustion. The S in the sample was converted to $SO_2$ and, along with N$_2$ and CO$_2$, was passed through two H$_2$O traps. The purified gases were then separated in a 30 cm, 0.5″ OD, QS GC column held at 45 °C and passed into the mass spectrometer—N$_2$ and CO$_2$ first, and then $SO_2$. Approximately one-third of the $SO_2$ was cracked to SO in the source, allowing $^{34}S$ to be measured by continuously monitoring masses 48, 49, and 50. The mass peaks were plotted, the area under each mass peak was determined, and the isotope ratios were calculated. These ratios were referenced to ratios determined on in-house reference materials analyzed in the same analytical run. The raw ratio data were corrected for drift over the course of the run along with blank/linearity effects, and if present,

then normalized to the international standards. Analytical precisions, based on the replicate analyses of international reference materials, were $\pm 0.3\permil$ for $\delta^{34}S$.

### 2.3.3. Compartmental Analysis to Calculate the Influx and Efflux of Sulfate

The $SO_4^{2-}$, total S, and $\delta^{34}S$ ratio data were used in a compartmental analysis of a single pool system for using stable isotope tracer data [56,57] modified from models using radioisotope data for ion uptake by fish and crayfish [58–60]. Compartmental analysis models the movement of a solute between two compartments based on diffusion and mass conservation. In this system, we are measuring the movement of $SO_4^{2-}$ between the surrounding water and the animal across semipermeable epithelial cell membranes that may be facilitated by the presence of $SO_4^{2-}$-specific transporter proteins. This is because the $SO_4^{2-}$ in the test water can be traced with the artificially elevated levels of $^{34}S$ by the $SO_4^{2-}$-influx rate ($J_{in}$). However, because only the change in the $SO_4^{2-}$ concentration in the test waters was used to measure the net $SO_4^{2-}$-flux rate ($J_{net}$), the $SO_4^{2-}$-efflux rate ($J_{out}$) may include S from sources other than the test water, such as food.

### 2.3.4. Calculations

$J_{net}$ was measured as the change in the water $SO_4^{2-}$ concentration during the exposure period ($t = 1$ day):

$$J_{net} = \frac{\left[SO_4^{2-}\right]_0 \times V_0 - \left[SO_4^{2-}\right]_t \times V_t}{M \times t} \tag{1}$$

where $[SO_4^{2-}]_0$ and $[SO_4^{2-}]_t$ are the concentrations of $SO_4^{2-}$ in the water ($\mu mol\ L^{-1}$) at the beginning and end of the exposure period, respectively; $V$ is the volume of the water in the chamber (L) measured at $t = 0$ and $t \approx 24$ h to account for evaporation; $M$ is the mass (g) of the animal placed into the container; and $t$ is the length of the exposure period (days).

$J_{in}$ was measured as the changes in the fractional molar abundance of $^{34}S$ and total S concentration in the animal during the exposure interval relative to the initial fractional molar abundance of $^{34}S$ in the test waters:

$$J_{in} = \frac{\left[X\left(^{34}S\right)_{int(t)} \times [S]_{int(t)}\right] - \left[X\left(^{34}S\right)_{int(0)} \times [S]_{int(0)}\right]}{[X(^{34}S)_{bath} \times t]} \tag{2}$$

where $X(^{34}S)_{int(0)}$ is the initial fractional molar abundance of $^{34}S$ in the animal, $[S]_{int(0)}$ is the initial concentration of S in the animal ($\mu mol\ g^{-1}$), $X(^{34}S)_{int(t)}$ is the fractional molar abundance of $^{34}S$ at the end of the exposure, $[S]_{int(t)}$ is the concentration of S in the animal at the end of the exposure ($\mu mol\ /g^{-1}$), $X(^{34}S)_{bath}$ is the fractional molar abundance of $^{34}S$ in the test waters, and $t$ is the length of the exposure period (day).

The fractional molar abundance of $^{34}S$ or $X(^{34}S)$ for a sample is calculated from the molar ratio of $^{34}S$ to $^{32}S$ or $R(^{34}S/^{32}S)$ for the sample by:

$$X\left(^{34}S\right) = \frac{R\left(^{34}S/^{32}S\right)}{1 + R(^{34}S/^{32}S)} \tag{3}$$

and $R(^{34}S/^{32}S)$ is calculated from the reported $\delta(^{34}S/^{32}S)$ by:

$$R\left(^{34}S/^{32}S\right) = \frac{\delta\left(^{34}S/^{32}S\right) \times N\left(^{34}S\right)_{std}/N\left(^{32}S\right)_{std}}{1000} + N\left(^{34}S\right)_{std}/N\left(^{32}S\right)_{std} \tag{4}$$

where $N(^{34}S)_{std}/N(^{32}S)_{std}$ is the molar ratio of the heavy stable isotope in the standard material, Canyon Diablo Troilite, which by convention is assigned a $N(^{34}S)/N(^{32}S)$ of 0.045005 [61]. The values are divided by 1000, because $\delta(^{34}S/^{32}S)$ is reported in parts per mille relative to the standard material.

$J_{out}$ is calculated as the difference between $J_{net}$ and $J_{in}$ [58]:

$$J_{out} = J_{net} - J_{in} \tag{5}$$

### 2.3.5. Statistical Analysis

As the measurements of $SO_4^{2-}$ and $\delta(^{34}S/^{32}S)$ in the water and in animals at the beginning of the exposures were made on and summarized for replicate subsamples, the variation associated with these measurements was pooled as appropriate with the variation of the calculated variables, $J_{in}$, $J_{out}$, and $J_{net}$. Then, the calculated variables for each species were tested to determine whether each variable was significantly different from 0 using a t-test (PROC TTEST, SAS Institute, Cary, NC, USA). Because six variable-by-concentration combinations were tested for each species, a Bonferroni adjustment of p = 0.0083 was used.

## 3. Results

The measured $SO_4^{2-}$ concentrations in the artificial water used with the different species were variable compared to the target concentrations (Table 2). The measured $\delta(^{34}S/^{32}S)$ for the artificial waters (Table 2) were more than 100 times the initial measured $\delta(^{34}S/^{32}S)$ of the animal tissues, which were +11.386 ± 0.090, −1.420 ± 0.910, −2.304 ± 0.987, and −4.617 ± 0.530 for the fathead minnows, paper pondshells, red swamp crayfish, and two-lined mayflies, respectively. The summary statistics for all the variables used in the compartmental analyses may be found in Table S1.

**Table 2.** Measured initial characteristics of $SO_4^{2-}$ in the test water for each species exposure. The column headings are the nominal concentrations of $SO_4^{2-}$. The values are the sample mean ± 1 standard error. $[SO_4^{2-}]$ (mmol L$^{-1}$) is the concentration of $SO_4^{2-}$, and $\delta(^{34}S/^{32}S)$ (‰) is molar ratio of $^{34}S/^{32}S$ of the $SO_4^{2-}$.

| Variable | Species | 0.49 mmol L$^{-1}$ | | 1.23 mmol L$^{-1}$ | |
|---|---|---|---|---|---|
| | | **n** | **Value** | **n** | **Value** |
| $[SO_4^{2-}]$ (mmol L$^{-1}$) | Fathead minnow | 5 | 0.52 ± 0.03 | 5 | 1.67 ± 0.03 |
| | Paper pondshell | 5 | 0.35 ± 0.01 | 5 | 0.98 ± 0.06 |
| | Red swamp crayfish | 10 | 0.45 ± 0.01 | 10 | 1.15 ± 0.01 |
| | Two-lined mayfly | 1 | 0.48 | 3 | 1.19 ± 0.04 |
| $\delta(^{34}S/^{32}S)$ (‰) | Fathead minnow | 5 | +1377.6 ± 1.3 | 5 | +1433.9 ± 0.8 |
| | Paper pondshell | 5 | +1432.7 ± 34.1 | 5 | +1452.6 ± 0.5 |
| | Red swamp crayfish | 5 | +1475.6 ± 2.8 | 5 | +1461.3 ± 1.0 |
| | Two-lined mayfly | 5 | +1411.6 ± 3.0 | 5 | +1365.7 ± 4.4 |

The means of $J_{in}$ for $SO_4^{2-}$ were positive for all the species and ranged from 2.14 to 13.32 μmol g$^{-1}$ day$^{-1}$ among the species and two nominal sulfate concentrations (Figure 1). The $J_{in}$ in 5 of 50, 12 of 50, 13 of 50, and 8 of 39 individual exposures of the fish, mollusk, crayfish, and mayfly were negative, and there were large confidence bounds around the means (i.e., the coefficient of variation ranges from 0.54 to 3.63). In part, notable variance was added to the means of $J_{in}$ because of unexpected variation in the measurements of the initial animal S concentrations (Table 3), but $J_{in}$ was significantly greater than 0 at both target $SO_4^{2-}$ concentrations in the fish and mollusk and at the lower $SO_4^{2-}$ concentration (i.e., 0.49 mmol L$^{-1}$) in the crayfish (Table 4). Additionally, $J_{in}$ increased between the two $SO_4^{2-}$ concentrations in the fish and mollusk (Figure 1), although the increase was statistically significant only for the fish (df = 48, $t$ = 2.70, p = 0.009) and the mollusk (df = 48, $t$ = 1.13, p = 0.20).

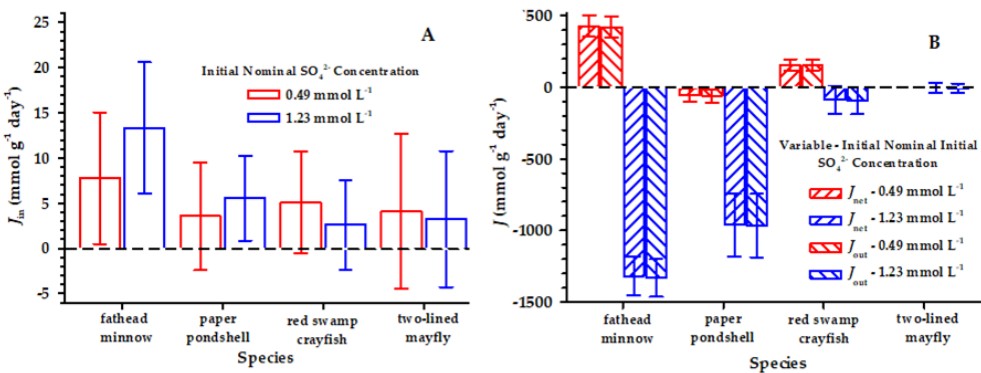

**Figure 1.** Mean $SO_4^{2-}$ influx ($J_{in}$) (**A**), efflux ($J_{out}$), and net flux ($J_{net}$) (**B**) ($\mu$mol g$^{-1}$ day$^{-1}$) for each animal species—the fathead minnow, paper pondshell, red swamp crayfish, and two-lined mayfly—at the two target $SO_4^{2-}$ concentrations of 0.49 mmol L$^{-1}$ and 175 mmol L$^{-1}$. The error bars represent $\pm1$ standard error. Because of some mortality occurring in the mayfly experiment, the numbers of replicates were 17 and 22 for the 0.49- and 1.25-mmol L$^{-1}$ $SO_4^{2-}$ exposures, respectively. Moreover, an error in the initial water $SO_4^{2-}$ analysis in the mayfly experiment reduced the number of valid measurements for the initial 0.49 mmol L$^{-1}$ water $SO_4^{2-}$-concentration to 1, precluding a valid mean of $J_{out}$ and $J_{net}$ for the 0.49 mmol L$^{-1}$ $SO_4^{2-}$ exposure.

**Table 3.** Proportion of variance in the J value means contributed by the variances of the mean values for component variables. The proportions of variance for other component variables were <0.001, or means were not used in the calculations. $[S]_{int(0)}$ is the mean calculated initial S concentration ($\mu$mol g$^{-1}$) of each animal, and $[SO_4^{2-}]_0$ is the measured concentrations of $SO_4^{2-}$ in the water ($\mu$mol L$^{-1}$) at the beginning of the exposure. $J_{in}$, $J_{out}$, and $J_{net}$ are the influx rates, efflux rates, and net flux of $SO_4^{2-}$ ($\mu$mol g$^{-1}$ day$^{-1}$), respectively, for each species.

| Species | Target $SO_4^{2-}$ Concentration | J value | Variable | Proportion of Variance |
|---|---|---|---|---|
| Fathead minnow | 0.49 mmol L$^{-1}$ | $J_{in}$ | $[S]_{int(0)}$ | 0.603 |
| | | $J_{out}$ | $[SO_4^{2-}]_0$ | 0.089 |
| | | | $[S]_{int(0)}$ | 0.015 |
| | | $J_{net}$ | $[SO_4^{2-}]_0$ | 0.090 |
| | 1.23 mmol L$^{-1}$ | $J_{in}$ | $[S]_{int(0)}$ | 0.584 |
| | | $J_{out}$ | $[SO_4^{2-}]_0$ | 0.583 |
| | | | $[S]_{int(0)}$ | 0.006 |
| | | $J_{net}$ | $[SO_4^{2-}]_0$ | 0.058 |
| Paper pondshell | 0.49 mmol L$^{-1}$ | $J_{in}$ | $[S]_{int(0)}$ | 0.172 |
| | | $J_{out}$ | $[SO_4^{2-}]_0$ | 0.046 |
| | | | $[S]_{int(0)}$ | 0.021 |
| | | $J_{net}$ | $[SO_4^{2-}]_0$ | 0.047 |
| | 1.23 mmol L$^{-1}$ | $J_{in}$ | $[S]_{int(0)}$ | 0.519 |
| | | $J_{out}$ | $[SO_4^{2-}]_0$ | 0.059 |
| | | | $[S]_{int(0)}$ | 0.001 |
| | | $J_{net}$ | $[SO_4^{2-}]_0$ | 0.059 |
| Red swamp crayfish | 0.49 mmol L$^{-1}$ | $J_{in}$ | $[S]_{int(0)}$ | 0.582 |
| | | | $[SO_4^{2-}]_0$ | 0.028 |
| | | $J_{out}$ | $[S]_{int(0)}$ | 0.012 |
| | | $J_{net}$ | $[SO_4^{2-}]_0$ | 0.072 |
| | 1.23 mmol L$^{-1}$ | $J_{in}$ | $[S]_{int(0)}$ | 0.747 |
| | | | $[SO_4^{2-}]_0$ | 0.002 |
| | | $J_{out}$ | $[S]_{int(0)}$ | 0.002 |
| | | $J_{net}$ | $[SO_4^{2-}]_0$ | 0.005 |

**Table 3.** *Cont.*

| Species | Target $SO_4^{2-}$ Concentration | $J$ value | Variable | Proportion of Variance |
|---|---|---|---|---|
| Two-lined mayfly | 0.49 mmol L$^{-1}$ | $J_{in}$ | $[S]_{int(0)}$ | 0.890 |
| | 1.23 mmol L$^{-1}$ | $J_{in}$ | $[S]_{int(0)}$ | 0.941 |
| | | $J_{out}$ | $\left[SO_4^{2-}\right]_0$ | 0.346 |
| | | | $[S]_{int(0)}$ | 0.084 |
| | | $J_{net}$ | $\left[SO_4^{2-}\right]_0$ | 0.380 |

**Table 4.** T-value (df and p) for the test of the $H_0$ that each mean $J = 0$. m$J_{in}$, m$J_{out}$, and m$J_{net}$ are the mean influx rate, mean efflux rate, and mean net flux of $SO_4^{2-}$, respectively, for each species. Because of the multiple tests for each animal species, the Bonferroni adjustment of $p$ is 0.0083. The *t*-values that indicate a mean J value statistically different from 0 are in bold.

| Species | Target $SO_4^{2-}$ Concentration | $H_0$: m$J_{in} = 0$ *t*-Value (df, p) | $H_0$: m$J_{out} = 0$ *t*-Value (df, p) | $H_0$: m$J_{net} = 0$ *t*-Value (df, p) |
|---|---|---|---|---|
| Fathead minnow | 0.49 mmol/L | **5.32 (24, <0.0001)** | **27.88 (24, <0.0001)** | **28.60 (24, <0.0001)** |
| | 1.23 mmol/L | **9.15 (24, <0.0001)** | **−49.38 (23, <0.0001)** | **−48.51 (23, <0.0001)** |
| Paper pondshell | 0.49 mmol/L | **2.98 (24, 0.0066)** | **−6.38 (24, <0.0001)** | **−6.04 (24, <0.0001)** |
| | 1.23 mmol/L | **5.91 (24, <0.0001)** | **−21.78 (24, <0.0001)** | **−21.74 (24, <0.0001)** |
| Red swamp crayfish | 0.49 mmol/L | **4.50 (24, 0.0001)** | **19.26 (24, <0.0001)** | **20.34 (24, <0.0001)** |
| | 1.23 mmol/L | 2.60 (24, 0.0155) | **−4.80 (24, <0.0001)** | **−4.68 (24, <0.0001)** |
| Two-lined mayfly | 0.49 mmol/L | 1.95 (17, 0.0680) | | |
| | 1.23 mmol/L | 1.97 (21, 0.0623) | −0.33 (2, 0.77) | −0.16 (2, 0.89) |

The means of $J_{out}$ and $J_{net}$ were much more variable but suggested a net excretion of $SO_4^{2-}$ by the four species (Figure 1, Table 4). Part of this variation was the result of unexpected variation in the measurements of the initial water concentrations of $SO_4^{2-}$ (Table 3).

## 4. Discussion

Because we measured an increase in whole animal $R(^{34}S/^{32}S)$ in animals exposed to water where the added $SO_4^{2-}$ was highly enriched with $^{34}S$, $J_{in}$ measured the uptake of $SO_4^{2-}$ from the water. Presumably, the $SO_4^{2-}$ moved through $SO_4^{2-}$-transporters on external epithelial membranes, such as the gills, chloride cells, or integument, because other ions commonly move through epithelial membranes via various intercellular pathways involving various ion transporters. Although our methods used whole animal assays and therefore did not definitively distinguish between external surfaces and internal tissues, the measured $J_{in}$, when expressed in the same units, are of a similar range of magnitude as the measurements of the uptake of other ions, including Cl$^-$ anions, at similar water concentrations [62–65]. Even though our methods do not distinguish between ionocyte-mediated transport and paracellular transport, which has been described for other ions, paracellular transport does not generally occur against concentration gradients, and we expect that it would take longer than 24 h for the equilibration of the isotope between the test solution and the internal milieu without the aid of facilitated or active transport [66–69].

The larger $J_{out}$ suggests that there is an internal pool of $SO_4^{2-}$ supplied by sources in addition to uptake from the water, such as from food [70,71]. By measuring the change in the $SO_4^{2-}$ concentration in the water, $J_{out}$ and $J_{net}$ measure all the effluxes of $SO_4^{2-}$ between the animals and the water, including renal excretion.

A number of other studies have observed $SO_4^{2-}$ transporters in aquatic animals, primarily associated with internal epithelial membranes. In teleost fish, two types of epithelial $SO_4^{2-}$ transporters, a Na$^+$/$SO_4^{2-}$-cotransporter (NaS1, SLC13 family) and a $SO_4^{2-}$/anion-exchanger (Sat1, SLC26 family), have been identified in the proximal tubules of the kidneys that are primarily involved in $SO_4^{2-}$ excretion and resorption [35,72,73]. The functional analysis of freshwater mollusks suggests a similar renal regulation of $SO_4^{2-}$ [26,30]. In mosquito (*Aedes campestris*) larvae from saline lakes, transporters

associated with the Malpighian tubules excrete $SO_4^{2-}$ [74,75]. In marine Atlantic lobsters (*Homarus americanus*), $SO_4^{2-}$ transporters associated with hepatopancreatic epithelia excrete $SO_4^{2-}$ in exchange for either $C_2H_4^{2-}$ or $Cl^-$ [76–79]. The $Na^+/SO_4^{2-}$-cotransporter, NaS1, was also detected in the intestinal tissue of zebrafish (*Danio rerio*) but not gill tissues [72]. However, our experiments were not designed to identify the types of $SO_4^{2-}$ transporters beyond their role in uptake from the ambient water.

The amino acid cysteine and S-containing proteins are synthesized in metazoans from the amino acid methionine. While the initial step is $SO_4^{2-}$ activation to PAPS, S is added to methionine by the sulfate assimilatory reduction of PAPS, a pathway not found in metazoans [40–43]. Therefore, these amino acids are sources of S in animals via ingestion. In other biomolecules, sulfate groups are transferred from PAPS by a sulfonation pathway, which is found in metazoans [41,43]. Therefore, inorganic $SO_4^{2-}$ from some source is required by animals. The renal reabsorption of $SO_4^{2-}$ is likely part of the source, but the source is also supplied by the direct uptake of $SO_4^{2-}$ from the water, which we observed in our experiments.

In freshwater nymphs of Plecoptera, Ephemeroptera, and Trichoptera, Scheibener et al. [28] measured the uptake of $SO_4^{2-}$ and found that this uptake was inhibited by increased $Na^+$ water concentrations, suggesting the presence of a $Na^+/SO_4^{2-}$-cotransporter. Buchwalter et al. [29] identified similar $SO_4^{2-}$ transporters in the mayfly, *Neocloeon trangulifer*, but the localization of these transporters was not determined. This research also observed that $SO_4^{2-}$ uptake increased with an increasing $SO_4^{2-}$ water concentration.

In conclusion, our study, along with studies from the Buchwalter laboratory [28,29], suggests that there is direct uptake of $SO_4^{2-}$ from the water in these four groups of freshwater animals. Additionally, there is some evidence that this uptake may increase with the water concentration of $SO_4^{2-}$. However, the uptake of $SO_4^{2-}$ from the water is not the only source of S, and S from food likely contributes to the $SO_4^{2-}$ excreted by these animals. Therefore, elevated water $SO_4^{2-}$ may interact with other ions to have ionoregulatory effects in freshwater animals that could cause the effects observed by more traditional ecotoxicological studies.

A next step for research on any of these freshwater animals would be to sequence, locate, and functionally characterize any $SO_4^{2-}$-transporters on their gills or other external epithelial membranes, as has been done for other ions [80–83]. Such information would clarify the potential interactions between $SO_4^{2-}$ and other ions, such as $Na^+$, when these ions are elevated in freshwaters. This will more completely identify the potential pathways for adverse outcomes [84] for elevated $SO_4^{2-}$ in freshwaters and better support risk assessments, leading to the development of water-quality benchmarks or criteria.

**Supplementary Materials:** The following are available online at http://www.mdpi.com/2073-4441/12/5/1496/s1, Table S1: Summary statistics (mean, standard error (SE), and n for all variables used in the equations to estimate $J_{in}$, $J_{out}$, and $J_{net}$ ($\mu M\ g^{-1}\ day^{-1}$).

**Author Contributions:** Conceptualization, M.B.G. and J.M.L.; methodology, M.B.G., J.M.L. and H.H.; validation, M.B.G. and H.H.; formal analysis, M.B.G.; investigation, M.B.G., J.M.L. and H.H.; resources, M.B.G. and J.M.L.; data curation, M.B.G. and H.H.; writing—original draft preparation, M.B.G. and H.H.; writing—review and editing, M.B.G. and J.M.L.; visualization, M.B.G. and supervision, M.B.G. and J.M.L.; All authors have read and agreed to the published version of the manuscript.

**Funding:** This study is based on work supported by the USEPA.

**Acknowledgments:** M. A. McGregor and staff of the Kentucky Department of Fish and Wildlife Resources' Center for Mollusk Conservation (Frankfort, KY) cultured and supplied us with the paper floater adults. W. Thoeny (Pegasus Technical Services, Cincinnati, OH) assisted in the laboratory. The modification of the calculations for use with sulfur stable isotope data were reviewed by C. Wood (University of British Columbia, Vancouver, BC, Canada). Earlier drafts of this manuscript were reviewed and improved by comments from K. Fritz, S. Cormier (U.S. Environmental Protection Agency, Office of Research and Development, Cincinnati, OH) and three anonymous peer reviewers. This manuscript was prepared by USEPA, ORD, National Center for Environmental Assessment, Cincinnati Division and National Exposure Research Laboratory, Systems Exposure Division. It has been subjected to the agency's peer and administration review and approved for publication. However, the views expressed are those of the authors and do not necessarily represent the views or policies of the USEPA. Any use of trade, firm, or product names is for descriptive purposes only and does not imply endorsement by the USEPA.

**Conflicts of Interest:** The authors declare no conflict of interest.

**Availability of Data:** The compiled data set used in these analyses can be viewed or downloaded from USEPA's Science Hub at https://doi.org/10.23719/1504445.

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
