# Peer review of "Uptake of Sulfate from Ambient Water by Freshwater Animals"

_water, doi:10.3390/w12051496_

Round 1

Reviewer 1 Report

Water-779119

Recommendation: Reject

Full Title: Uptake of sulfate from ambient water by freshwater animals

Comments to Author

The manuscript by Griffith et al. entitled ‘Uptake of sulfate from ambient water by freshwater animals’ examines a sodium sulfate (i.e., sulfur radioisotope) uptake across four common aquatic species to determine whether sulfate uptake occurs across multiple animals. All aquatic organisms were exposed to two different sulfate artificial water levels in which sodium sulfate was enriched with the 34S isotope. Sulfate influx occurred across all organisms studied; however, net sulfate flux per species was inconsistent, with the bivalve paper floater organism demonstrated to actively excrete sulfate across both studied concentrations and was positively associated with increasing sulfate levels.

While the research subject fits with the overall journal focus, the manuscript requires restructuring, major justifications and clarifications for the experimental methods before its ready for publication. Overall the entire manuscript requires adjustments to enhance flow, transitions between paragraphs, and drastically increased clarity on the overarching research goal. Several grammatical errors (e.g., run on sentences, typos) need to be fixed. Proper paragrah structure is lacking while multiple sentences can be combine and/or removed to improve clarity and enhance flow. Examples of such errors and others are provided within minor comments. Too many items within sections need clarification, the writing style needs major adjustments, and the results aren't contextualized within the discussion and thus my suggestion is to reject the manuscript.

Minor Comments

Major Comments

The introduction sets up the importance of considering sulfate as an overall contributor for “total water ionic strength or salinity” toxicity to freshwater organisms; however, the present results extrapolated to the current knowledge on the topic falls quite short in the discussion. The results were not mentioned in that very same valuable context, as mentioned in the introduction, within the discussion. Instead, statements about measured sulfate levels in other freshwater organisms and the physiological role of sulfur in animals were discussed, but not in the context of linking the present experimental results to those statements. Discussing the clear or not so apparent species differences in sulfate flux within organisms is needed in addition to the implications of the demonstrated sulfate uptake toward freshwater aquatic organism toxicity. A brief mention of sulfate’s role in metal-complexation and how chromium and molybdenum mimic sulfate would also be relevant in the context of its importance in ecotoxicology. Overall additional contextualization of the results to the current state of knowledge and hazard paradigm concerning sulfate toxicity is necessary.

Line 41-44: Break into two sentences.

Line 66-70. Break into two sentences.

Line 78-80: Strengthen this short paragraph by clearly stating the research objectives and include the overarching rationale for the uptake study. What are the expected results, how will they contribute to understanding the contribution of sulfate to “ionic” toxicity, and which aquatic freshwater species are thus expected to be at risk?

L83: Is this a typo? Developmental stage will have a pretty big influence I would think, even if only on the basis of body size. Therefore can the author justify this statement further?

Line 88-120: This particular methods section can be written more concisely with a different verb tense. Instead of “We used fathead minnows from…” rephrase to something to the likes of “Fathead minnows used in the present study were cultured at X lab according to standard USEPA husbandry procedures.”

Line 110: mMol L-1 isn’t a correct unit and must be changed here and throughout the entire manuscript

Line 123: replace double and quintruple with “2X” and “5X” or something else similar to typical scientific writing.

Line 124: This seems to suggest the author tested adsorption, which is contradictory to other statements. Animals were blotted dry, which should reduce adsorption (e.g., to mucus), but this sentence implies they aren’t sure that this worked. Adsorbed sulfate should not be counted as absorbed sulfate, and will confound the data significantly. Please clarify.

Line 135: As written, it cannot be understood… could not use the test water? Another typo? Please clarify and reword this sentence.

Line 141: How many animals per chamber? Could be pseudoreplication.

Line 148-149: How come the sulfur isotope wasn’t measured at 24 hours?

Line 153: What can you really tell from animals that were clearly unhealthy. This is a big red flag, and given that the mayfly experiment end up with n = 1 anyway for one of the sulfate concentrations, the mayfly data should be deleted. I wouldn’t trust physiological measurements in dying animals.

Line 198-201: Be sure to define “t” in this equation (i.e., 24h = duration of exposure)

Line 207: typo… “parts per mill relative”

Line 218-222: How do you get negative numbers for the initial artificial water:organism tissue enrichment ratio? Please clarify.

Line 223: By stating “mean estimates” its interpreted as such values were not calculated, which is false. You calculated the mean Jin thus delete the word “estimate” in this sentences and all others to follow to increase clarity.

Line 227-228: Are the initial animal S concentrations so variable because of such a short “acclimation” duration in which animals were not able to reach a steady state internal sulfate concentration? Also, the actual experiment duration of 24 hours is also very short. Please justify both durations and/or clearly clarify the implications of your results in the confines of the experimental methodology.

Looking at the graph where the SEMs are so huge it is almost impossible to see how any of these data are significantly different from 0, with perhaps the exception of the high FHM. I am very skeptical. Did the authors confirm that the data could be interrogated by parametric statistics?

Line 234: Are four decimal places necessary for values in Table 2?

Line 239: Each graph in Figure 1 should be labeled with A and B to explicitly refer to each graph accordingly. Graphs need to be reformatted to minimize the x-axis text. For example, make a double bar for a single x-axis tick per animal and color-code the bars to represent the two different sulfate concentrations.

Line 246: The last half of the discussion is really weak, and contains all material that belongs in the introduction. As a simple rule if there is no or very little integrated reference to the collected data then it belongs in the introduction.

Line 263: The numbers in Table 4 do not add up here. Jnet should equal Jin – Jout (or vice versa), but this is clearly not the case in the table, so it is really confusing.

I cannot reconcile how you get a positive Jout (which is theoretically the same as a Jin), but then when you actually measure Jin you get a number really close to zero. Isn’t that inconsistent?

Line 331: Reference formatting is consistent. Thank you

Reviewer 2 Report

Review of Water-779119: “Uptake of sulfate from ambient water by freshwater animals” by Griffith MB, Lazorchak JM, and Haring H. 

This manuscript describes the results of sulfate flux experiments with four freshwater animals: a fish, a mussel, a crayfish, and a mayfly.  I believe the author does a good job of explaining the need for the study, and the methods are for the most part well described.  Since only one type of experiment was conducted with four different species, the technical aspects of the paper are straightforward and sound.  The main, relatively minor criticism I have of the paper is in the packaging.  Specifically, while a very large number of previous studies are cited, and the authors delve deeply into the finer physiological details, what is missing is a better statement on what this all means.  For example, in the introduction the authors discuss regulatory needs and implications, but then do not circle back in the discussion to how the results of this study might inform regulations.  Also, there are two other recent papers out of David Buchwalter’s lab on sulfate fluxes in aquatic insects and how those relate to toxicity that are only cited in passing in the last paragraph of the paper.  I think more comparison with those results might be warranted.  Overall, I feel like these are issues that can be overcome and the paper should be acceptable for publication with attention to those details.

Specific comments follow:

Abstract

Page 1, line 16. “paper floater” -  While it is possible to find references to “paper floater”, this species is most commonly referred to as “paper pondshell”.  (though common names  probably do not matter much)

Introduction

Page 1, line 30.  Most texts use either “specific conductance” or “conductivity” not “specific conductivity”.  Also, is SC not a physical measurement rather than a chemical measurement?

Page 1, lines 33 and 34-  I’m not sure I agree that this is what the cited papers [3-6] were advocating.  Conversely, at the time those papers were published, the conventional wisdom was that anions were responsible for salt toxicity so the cited studies reported toxicity data in terms of anion concentrations. 

Page 2, line 42 –Regarding minimal influence of Na, it is important to acknowledge here that Mount et al (2016) refute this.

Page 2, lines 53-56 – David Buchwalter's lab has a couple papers on sulfate flux in aquatic insects and should be included here.  These are probably the most similar studies to the present one.

Page 2, lines 57-77 - This paragraph has some extended sentences that make it more difficult to find the point.  I suggest breaking up some of the sentences into shorter thoughts.

Materials and Methods

Page 3, line 100 – “about 15 cm of water”… use a volume measure rather than a length measure?

Page 3, line 102 –Where could one obtain crayfish food?  And how much were they fed?

Page 3, line 105 – “green algal culture”… more detail is needed.  What kind of algae; where did you get it; is 30,000 cells/ml the final concentration in the tank or the concentration of the stock?

Page 3, line 108 “grain flake food”… as above, more detail is needed.

Page 3, line 126 - Soucek and Dickinson (2015) reported a SO4 EC20 of 1.5 mM/L (145 mg/L).  Not too far off.

Table 1 – Why are salt concentrations reported in mg/L whie ion concentrations are reported in mM/L

Page 4, line 145 – Can the specific minimum mass be provided?

Page 4, line 148 – Do I read correctly that isotope ratios were not measured in 24-h samples?  Is it not important to have this info?

Page 5, lines 185 to 188 - I think that for the unfamiliar reader, more detail is needed on exactly what compartmental analysis is and why it was done.

Discussion

In general, I think the discussion would greatly benefit from either an opening or a concluding paragraph summarizing the overall picture of what was found.  Furthermore as discussed above, placing the results in the context of 1) what this might mean for toxicity, and 2) what it might mean to regulatory efforts…would be helpful.  This is brought up in the introduction, but not in the discussion which focuses on the physiological and cellular aspects.

For example, page 9, lines 285 to 294 – While the scope of literature review is impressive, I’m not sure this level of detail is necessary for this manuscript.

Reviewer 3 Report

The manuscript entitled “Uptake of sulfate from ambient water by freshwater animals” by Griffith et al., is an excellently written manuscript that showed clearly how the regulation of sulfate in freshwater animals is more regulated than we previously thought. It is very well described, written and the experimental design is excellent. The manuscript is totally acceptable for publication in its form, after doing some minor corrections that mostly are typos.

Minor changes/corrections:

Line 31: replace “on” with “about”.

Line 36: replace “hardrock” with “hard rock”.

Line 37: replace “transformed to” with “transformed into”.

Line 41: replace “assume” with “assumes”.

Line 59: add “the” before “continuous”.

Line 65: add “the” before “absorption”.

Line 84: replace “stage” with “stages”.

Line 89: replace “originally considered” with “considered originally”.

Line 109: replace “prior to use” with “before using”.

Line 136: add “of” after “because”.

Line 142: add a comma between “mayfly” and “respectively”.

Line 153: replace “whole body” with “whole-body”.

Line 212: add “the” before “variation”.

Line 227: add “a” before “significant”.

Line 231: replace “suggest” with “suggested”.

Line 236: add “the” before “molar”.

Line 241: add “of” after “because”.

Line 252: replace “do” with “did”.

Line 258: replace “were” with “was”.

Round 2

Reviewer 1 Report

Water-779119

Recommendation: Accept (minor revisions)

Full Title: Uptake of sulfate from ambient water by freshwater animals

Comments to Author-Version 2

The manuscript by Griffith et al. entitled ‘Uptake of sulfate from ambient water by freshwater animals’ has been significantly improved from the first version. See a couple minor edits and comments below.

Minor Comments

Line 99-100: Delete sentence starting with “Similar…” Such a sentence should be incorporated above within the introduction rather than in the paragraph describing the study objectives.

Line 272: New figures (i.e., Figure 1) look great!

Line 283-285: The added sentence states the same remark twice and needs revision to increase clarity.

Line 298-299: Start the discussion with an overarching statement reaffirming the goal/objective of the study…and then go into the main result of the study. This first sentence is poorly written and needs revising to increase clarity (both R(34S/32S) and SO4 are mentioned twice in the same sentence…).

Line 298-314: Again, start the discussion with an overarching statement reaffirming the goal/objective of the study…and then go into the main result of the study. This first discussion paragraph, as written, states two flaws and/or things not measure or distinguished in the paper…instead of mentioning positive results and overarching inferences from the study; therefore, revisions seem necessary.

Line 364: Typo. Should be “to” instead of “do”… Also, great to see a sentence about how the present results could be used in ecological risk assessment.
